# Graphene-Binding Peptide in Fusion with SARS-CoV-2 Antigen for Electrochemical Immunosensor Construction

**DOI:** 10.3390/bios12100885

**Published:** 2022-10-17

**Authors:** Beatriz A. Braz, Manuel Hospinal-Santiani, Gustavo Martins, Cristian S. Pinto, Aldo J. G. Zarbin, Breno C. B. Beirão, Vanete Thomaz-Soccol, Márcio F. Bergamini, Luiz H. Marcolino-Junior, Carlos R. Soccol

**Affiliations:** 1Molecular Biology Laboratory, Graduate Program in Bioprocess Engineering and Biotechnology, Federal University of Paraná (UFPR), Curitiba 81531-980, PR, Brazil; 2Laboratory of Electrochemical Sensors (LabSensE), Department of Chemistry, Federal University of Paraná (UFPR), Curitiba 81531-980, PR, Brazil; 3Materials Chemistry Group (GQM), Department of Chemistry, Federal University of Paraná (UFPR), Curitiba 81531-980, PR, Brazil; 4Graduate Program in Microbiology, Parasitology, and Pathology, Federal University of Paraná (UFPR), Curitiba 81531-980, PR, Brazil

**Keywords:** electrochemical immunosensor, solid-binding peptide (SBP), graphene, SARS-CoV-2

## Abstract

The development of immunosensors to detect antibodies or antigens has stood out in the face of traditional methods for diagnosing emerging diseases such as the one caused by the SARS-CoV-2 virus. The present study reports the construction of a simplified electrochemical immunosensor using a graphene-binding peptide applied as a recognition site to detect SARS-CoV-2 antibodies. A screen-printed electrode was used for sensor preparation by adding a solution of peptide and reduced graphene oxide (rGO). The peptide-rGO suspension was characterized by scanning electron microscopy (SEM), Raman spectroscopy, and Fourier transform infrared spectroscopy (FT-IR). The electrochemical characterization (electrochemical impedance spectroscopy—EIS, cyclic voltammetry—CV and differential pulse voltammetry—DPV) was performed on the modified electrode. The immunosensor response is based on the decrease in the faradaic signal of an electrochemical probe resulting from immunocomplex formation. Using the best set of experimental conditions, the analytic curve obtained showed a good linear regression (r^2^ = 0.913) and a limit of detection (LOD) of 0.77 μg mL^−1^ for antibody detection. The CV and EIS results proved the efficiency of device assembly. The high selectivity of the platform, which can be attributed to the peptide, was demonstrated by the decrease in the current percentage for samples with antibody against the SARS-CoV-2 S protein and the increase in the other antibodies tested. Additionally, the DPV measurements showed a clearly distinguishable response in assays against human serum samples, with sera with a response above 95% being considered negative, whereas responses below this value were considered positive. The diagnostic platform developed with specific peptides is promising and has the potential for application in the diagnosis of other infections that lead to high antibody titers.

## 1. Introduction

Biosensors are analytical devices that use biomaterials to detect analytes through biological reactions or interactions. Biomaterial-mediated reactions usually show key-lock selectivity, which is a huge advantage over other types of sensors. Therefore, these reactions have been applied as an analytical tool for many applications, e.g., biomedical, environmental, food safety, drugs, and diagnostic applications for infectious disease, among others [1,2]. Biosensors that use immunological reactions as a mechanism for recognizing a target, either biological or not, are classified as immunosensors. The classification of signal transduction is defined as electrochemical, optical, or piezoelectric [3]. Electrochemical immunosensors combine the selectivity of immune reactions with the high sensitivity of electrochemical sensors. This synergism has been successfully applied to detect different types of viruses, e.g., hepatitis B and C [4,5], influenza [6], dengue [7], Zika [8], HIV [9,10], coronavirus [11,12], hantavirus [13,14,15], and others.

In terms of overall performance, electrochemical immunosensors show some advantages over traditional analytical techniques such as immunoassays (ELISA), due to their low-cost, portability, miniaturization, easy use, and on-site monitoring, which preconize and sustain the development of point-of-care tests. However, the analytical performance of immunosensors still demands improvements in terms of sensitivity since lower detection limits imply early diagnosis and selectivity, which is the pillar that sustains the application of sensors aiming at diagnosis.

In order to overcome this challenge, nanomaterials such as graphene, gold nanoparticles, and carbon nanotubes have been integrated into immunosensor platforms. Among them, reduced graphene oxide has drawn attention over electrochemical biosensing due to its remarkable conducting properties, large surface area, and easy modification [16,17]. Additionally, new recognition receptors such as aptamers and specific peptides have been extensively developed for binding to specific epitopes and antibodies [1,18,19]. Peptides have excellent recognition properties, showing their potential for a variety of diagnostic purposes, as mentioned before. The synthesis of specific peptides seeks the development of more sensitive and selective bioreceptors with a simpler overall assembly. Moreover, the conjugation or immobilization of biomolecules onto the surfaces of nanomaterials, preserving their biological activity, is of huge significance and has lately been under the spotlight of biosensor applications.

A simple electrochemical biosensor was elaborated [12] based on a highly specific synthetic peptide to detect the SARS-CoV-2 spike protein, using screen-printed gold electrodes functionalized with the thiolated peptide. The interaction with the spike protein was verified by EIS, and the platform showed high sensitivity and reproducibility and a limit of detection of 18.2 ng mL^−1^. The simplicity of the sensing platform combined with an on-demand synthetic peptide enabled detection of other viruses. Immunosensors were obtained [20] by synthesis of ferrocene (Fc)-containing peptides with the sequence Fc-Glu-(Ala)_n_-Cys-NH_2_, to form self-assembled monolayers on gold and be attached to antibodies. The highest performance in detecting C-reactive protein was achieved by an immunosensor with the peptide with two alanines (*n* = 2), whose chain length seemed appropriate for an efficient electron transfer. Using electrochemical impedance-derived capacitive spectroscopy, the limit of detection was 240 pM. The designed peptides applied in immunosensor construction, which show fast response and sensing performance, can be promising for low-cost platforms for point-of-care diagnostics.

This work aimed to obtain a simple sensing platform with the possibility of being able to monitor the course of several viral diseases. For this purpose, we proposed to develop an electrochemical immunosensor based on solid-binding peptide (SBP) with affinity to graphene. The SBP conjugates the antibody-binding site with support-binding residues, which are specific and bind to nanomaterial surfaces. This effect is based on the high affinity between chemical groups within amino acid residues and solid surfaces, thus the exact amino acid sequence has considerable importance in the manner that SBP behaves [21]. As a case study proof, a graphene-binding peptide was synthesized in fusion with a specific SARS-CoV-2-antibody binding site (S protein), resulting in a stable and easy-to-handle chimeric molecule.

With regard to the diagnosis of SARS-CoV-2, testing is mostly performed by the RT-PCR assay. However, despite its high accuracy, accessibility is still a challenge to be overcome, as these assays are usually centralized in research centers and pharmaceutical laboratories and usually demand long time periods, specialized professionals, and specific equipment [18,22]. In parallel, serologic assays evaluate the immune response against the virus by detecting antibodies (e.g., IgG, IgM, and IgA). Classically, IgG production is monitored since it represents the long-term and lasting immune response. Furthermore, the IgG class is selective against a specific epitope with memory effect, allowing it to remain above detectable thresholds for months or even years. Such antibodies are essential to avoiding and mitigating recurrent infections in a community. Therefore, its monitoring is essential to establish the extent and duration of immunity against SARS-CoV-2, and the antibody status will afford identifying the ability of a population to contract or resist infection.

## 2. Materials and Methods

### 2.1. Materials and Reagents

All chemicals were analytical or high-purity grade. Potassium ferricyanide K_3_[Fe(CN)_6_] was used as a probe. Phosphate buffer saline at pH 7.4 was obtained from potassium phosphate monobasic (KH_2_PO_4_), disodium hydrogen phosphate (Na_2_HPO_4_), sodium chloride (NaCl) and potassium chloride (KCl). The bovine serum albumin (BSA) and the antibody against Yellow Fever (YFV—0.16 mg mL^−1^) were provided by the Molecular Virology Laboratory at the Carlos Chagas Institute—FIOCRUZ/PR. This study included serum from adult and healthy volunteers collected before the beginning of the pandemic. Serum samples were obtained from COVID-19 patients confirmed by at least two conventional laboratory tests for SARS-CoV-2: Lateral Flow and RT-qPCR in May 2020. The screen-printed electrodes (SPE), DRP 110 (Aux.:C; Ref.:Ag) were purchased from Metrohm.

The rGO dispersion in water was obtained following the methodology described by Li et al. [23] using 100 mL of GO dispersion (0.030 mg mL^−1^), which was placed in a reflux system under heating (95 °C) in oil bath. Then, a 26 µL volume of a previously prepared hydrazine dihydrochloride solution (N_2_H_4_·2HCl) 2.73 mol L^−1^ and 65 µL of an aqueous ammonium hydroxide solution (NH_4_OH) 28% were added to the dispersion. The system was left under stirring at 95 °C for 1 h. Finally, the resulting dispersion was sonicated in an ultrasound bath for 20 min to disperse the solids formed at the air–liquid interface during the reduction process and subsequently characterized.

The selected peptide [24] containing a specific sequence with affinity to graphene (EPLQLKM) in fusion with a specific SARS-CoV-2-antibody binding site (S protein) was chemically synthesized using cleavable resin as a solid support, allowing the production of soluble peptides in aqueous buffer [25]. The method employed solid-phase peptide synthesis (SPPS) using an automatic synthesizer (Intavis Bioanalytical Instruments, Köln, Germany) and Fmoc-protected amino acids.

### 2.2. Morphological and Structural Characterization

Scanning electron microscopy (SEM) images were obtained in a Mira FEG-SEM electron microscope (Tescan) with an accelerating voltage of 10 kV and coupled to an EDS detector (Oxford Instruments) for elemental analysis. Raman spectra were acquired with a Raman Confocal WITec alpha 300R microscope with a lateral resolution of 200 nm, a vertical resolution of 500 nm and a 532 nm Ar+ laser with a power of 0.3 mW. The Fourier-transform infrared spectroscopy (FT-IR) measurements were performed with a Vertex 70 Brucker using 32 scans ranging from 4000 cm^−1^ to 500 cm^−1^. The samples were previously homogenized with KBr pellets and dried.

### 2.3. Electrochemical Measurements

The cyclic voltammetry (CV), differential pulse voltammetry (DPV), and electrochemical impedance spectroscopy (EIS) measurements were carried out in a Potentiostat/Galvanostat Metrohm AUTOLAB PGSTAT204 with the FRA32 M module using the NOVA 2.1 software. The experiments were conducted in a 0.1 mol L^−1^ phosphate-buffered saline (PBS) solution at pH 7.4, with 1.0 mmol L^−1^ of (K_3_[Fe(CN)_6_]) as an electrochemical probe. The EIS measurements were performed at a frequency range of 0.1 Hz–100 kHz with an AC amplitude of 10 mV and a potential condition equivalent to the K_3_[Fe(CN)_6_] anodic peak.

### 2.4. Immunosensor Construction

First, the peptide was added to the rGO water suspension (1:10 ratio) and kept under stirring at a controlled temperature (4 °C) for at least 1 h. With regard to immunosensor construction, the first step was the drop casting of 5 μL of peptide and rGO mixture to WE, followed by drying at 37 °C for 1 h. Then, in order to block non-specific sites of the sensor and mitigate parallel interactions, 4 μL of BSA (1.0 mg mL^−1^) was added and incubated at 4 °C in a water-saturated atmosphere for 15 min. From this step onwards, every incubation was performed in a water-saturated atmosphere. Next, the electrodes were washed three times by immersion in PBS solution to remove excess and unbound biomolecules. Finally, the antibody against the SARS-CoV-2 S protein (AbS) was detected by the decrease in current intensity of the redox probe. This step was carried out by incubating 4 μL of AbS solution at WE surface for 1 h. Figure 1 shows the schematic illustration of immunosensor construction. Each step was evaluated by CV (−0.3 V to 0.6 V, 50 mV s^−1^, three cycles) and DPV (−0.4 V to 0.5 V, 5 mV s^−1^).

### 2.5. Analytical Curve, Selectivity, and Stability

The analytical curve was recorded after AbS incubation in PBS (0.1 mol L^−1^) with a concentration ranging from 80 ng mL^−1^ to 5.2 μg mL^−1^. Voltametric measurements (CV and DPV) were conducted in PBS 0.1 mol L^−1^ at pH 7.4 with 1.0 mmol L^−1^ of (K_3_[Fe(CN)_6_]) as an electrochemical probe.

Selectivity assays were performed with samples of AbS, IgG antibody against SARS-CoV-2 N protein (AbN), and IgG antibody against YFV disease (AbYFV) with concentrations of 1.55 μg mL^−1^, 1.32 μg mL^−1^, and 1.60 μg mL^−1^, respectively.

Voltametric response stability was evaluated by maintaining the immunosensors under refrigeration (4 °C) for 3 weeks. After each period, antibody (AbS) detection was carried out as described before.

### 2.6. Serum Sample Testing

Real positive and negative (IgG) human serum samples (10,000× diluted) were analyzed to evaluate the immunosensor’s behavior in complex biological samples. The serum samples were added to WE and incubated under refrigeration (4 °C) for 1 h, followed by washing in PBS solution. The electrochemical measurements were carried out using the same experimental conditions described previously.

### 2.7. Ethical Issues

The present study was carried out in accordance with the Brazilian National Research Ethics Commission (CEP-CONEP) and CNS resolution 196/96. Approval was granted by the Ethics Committee of the Federal University of Paraná under Process no. 3.954.835.

## 3. Results

### 3.1. rGO and Peptide Characterization

The rGO dispersion in water and the peptide-rGO mixture were characterized on the SPE by SEM. The typical graphene-like morphology with connected 2D platelets could be seen in the image of the modified electrode (Figure 2a) as well as the carbon surface from the electrode (Appendix A). However, peptide presence was not observed in the SEM images, making it impossible to clearly identify them.

The FT-IR data obtained for the mixture (peptide and rGO) confirmed peptide adsorption onto the rGO and consequently over SPE. The rGO spectrum (Figure 2b—black line) showed an angular deformation at 1627 cm^−1^ from H_2_O molecule, OH angular deformation from COOH (1384 cm^−1^), and C-O-C stretching (1201 cm^−1^). Peptide contribution (Figure 2b—red line) could be identified by bands at 1679, 1537, 1400, and 1207 cm^−1^. The bands at 1679 and 1537 cm^−1^ could be attributed to the vibration of the NH bending corresponding to the amide bond, confirming peptide adsorption onto rGO [26,27,28].

Raman spectroscopy provides information about the structural disorders, crystallization, defects, and quality of carbon materials. The Raman spectra obtained for rGO and peptide-immobilized rGO are shown in Figure 2c. Carbon-based (nano)materials could be characterized by two main bands: the G band at 1583 cm^−1^, which occurred due to C=C stretching, and the D band at 1355 cm^−1^, attributed to defects in the graphene structure [29,30]. The ratio between the D and G bands can be used to estimate the degree of defects in the sample. Therefore, an ID/IG ratio of 0.63 was obtained for rGO and 0.85 for the mixture with peptide, which could be indirect evidence of the peptide/rGO interaction.

### 3.2. Electrochemical Evaluation of Immunosensor Construction

The proposed immunosensor was investigated by the voltametric behavior of the redox probe K_3_[Fe(CN)_6_], as shown in Figure 3a, where SPE represents the electrode without any modification. Peptide concentration and the rGO mixture were previously evaluated from 0.55 ng mL^−1^ to 0.11 μg mL^−1^, and the lowest concentration (0.55 ng mL^−1^) was used in this study. After peptide and rGO drop casting, the redox probe showed a decrease in current intensity from 21.03 μA to 18.84 μA (Figure 3b) in relation to SPE, which could be related to functional groups on the surface of the peptide and rGO and the layers formed, which hindered electron transfer during the redox reaction [11,31]. The peptide adsorption process in carbon nanomaterials such as rGO is driven by non-covalent interactions of electrostatic and van der Waals forces [32,33].

Subsequently, BSA (4 μL) was added and incubated at 4 °C, followed by a washing step in PBS. This anchoring process was performed to block the remaining active sites, avoiding non-specific interactions between antibodies and the device. As a result, the immobilization process was evaluated by the decrease in the probe redox signal due to electroactive surface blocking. Since there are no sites/residues for an effective BSA binding, the interaction is merely adsorptive. Nevertheless, it mitigates interactions or parallel reactions that may occur on the electrode surface, which would result from adsorption. Lastly, AbS (1.55 μg mL^−1^) was detected against the peptide on SPE, inducing the formation of an immunocomplex evaluated by the decrease in the faradaic current (ΔI = 2.56 μA). This decrease results from the steric hindrance due to immunocomplex formation and acts by partially blocking the electroactive surface, disturbing the interaction of the redox probe [34,35].

The immunosensor assembly was also evaluated by EIS measurements, as shown in the Nyquist plots (Appendix A), indicating a relationship between the impedance (charge transfer resistance—R*_ct_*) and CV data (current intensity of the anodic peak—I_pa_ for each assembling step. An R_ct_ increase, calculated based on the electrochemical circle fit (Randles circuit), was verified after peptide and rGO drop casting, with 2.71 kΩ for SPE, 9.50 kΩ after drop casting and, thereafter, up to 29.03 kΩ with BSA incubation, behaving as a blocker molecule. Along with the CV data, these EIS results indicate a modification on the electrode surface according to each anchoring stage. The optimized procedure described above was achieved based on previous studies [13,14,15] and applied in the following studies.

### 3.3. Analytical Performance

The immunosensor response to AbS was analyzed with concentrations varying from 80 ng mL^−1^ to 5.2 μg mL^−1^ in PBS solution, as seen in Figure 4a. The results show a variation in current intensity (ΔI) between BSA and the AbS voltametric signal following the increase in AbS concentration. The analytical curve showed promising electroanalytical results, with a good linear regression (r^2^ = 0.913) and a limit of detection (LOD) of 0.77 μg mL^−1^ for AbS detection. Concentrations smaller than 0.3 μg mL^−1^ showed responses but also a lack of linearity. These results are comparable with those of studies reported in the literature (Table 1), such as the impedimetric screening of the antigen (SARS-CoV-2) by a 3D-printed graphene-based surface immunosensor, which showed a detection limit of 0.5 ± 0.1 μg mL^−1^ in buffered and human serum samples [36]. Despite the sensitivity, the overall performance of the immunosensor along with its short assembly time, with only two simple steps, make it a very promising device.

The matrix effect, characteristic of complex biological samples, may cause nonspecific interactions and influence the biosensor’s sensitivity and selectivity. Parallel interactions may cause false signals, either positive or negative, affecting data interpretation [37,38]. To evaluate the matrix effect, the immunosensor’s voltametric response (Figure 4b) was studied with three different AbS concentrations: 0.77, 1.55, and 3.09 μg mL^−1^ in negative human serum (10,000× diluted). For all concentrations tested, the results showed a decrease (at least 20%) in the probe redox signal, with more emphasis on the AbS concentration of 1.55 μg mL^−1^, which caused a 42% reduction. Therefore, the immunosensor performance was not damaged by the matrix effect.

**Table 1 biosensors-12-00885-t001:** Electrochemical biosensors applied in the detection of SARS-CoV-2: recognition sites, techniques, immunoassay time, and lowest detectable values.

Recognition Sites	Detection Technique	Immunoassay Time ^[a]^	Lowest Detectable Value ^[b]^	Ref.
rGO and specific viral antigens	EIS	4 h	S1 protein—2.8 fM RDB—16.9 fM	[11]
Peptide	EIS	15 min	18.2 ng mL^−1^	[12]
Anti-spike antibody	CV and EIS	45 min	20 μg mL^−1^	[39]
Aptamer-RBD	EIS	40 min	1.30 pM	[40]
Graphene-peptide	DPV	60 min	0.77 μg mL^−1^	This study

^[a]^ Time for analyte interaction; ^[b]^ Lowest concentration of the linear working range or lowest concentration reported in the study.

### 3.4. Selectivity and Stability of the Immunosensor

In addition to their chemical versatility and easy modification, specific peptides also become highly selective [12]. In order to verify the immunosensor’s selectivity aiming to avoid false positive diagnostics, the platform was tested against AbYFV, AbN, and a blend of AbS with antibodies cited before (AbBlend), all in equivalent concentrations. The results obtained for the antibodies tested are shown in Figure 5. The increase observed in the current percentage of the AbYFV and AbN samples was possibly due to the desorption of the selective layer, not representing a proper interaction of the antibodies within the surface. On the other hand, the samples fortified with AbS showed decreased current percentages, proving an AbS–peptide interaction at the electrode and thus providing a positive diagnosis. Therefore, the proposed immunosensor showed excellent selectivity against the evaluated antibodies.

Nanomaterials used to develop immunosensors such as rGO enhance anchoring processes and provide a more biocompatible surface, maintaining highly stable biomolecules. Over time, this stability is essential for immunosensors since any degradation could affect the device’s performance [3,41]. The immunosensors were prepared until the BSA incubation step, after which they were stored at 4 °C. Sequentially, voltametric measurements were carried out after each period (Appendix A) against AbS (1.55 μg mL^−1^). This study was conducted with the immunosensor for the final readout. Therefore, a decrease in the current intensity of the device as a whole was observed over time. However, the developed immunosensor showed a positive response against AbS until the fifteenth day.

### 3.5. Immunosensor’s Testing against Serum

The immunosensor developed using a peptide with a specific sequence as a recognition site was evaluated against negative and positive human serum samples, both with a 10,000× dilution factor. Since the sensor is label-free, it can detect any type of antibody (IgG, IgA, and IgM) capable of interacting with the peptide. Therefore, sample concentration becomes a sum of all antibodies. As a result, the concentration must be very high, even though this is a subjective parameter that depends on each individual and does not represent the pathophysiological range, i.e., the severity of the disease, but rather an immunogenicity parameter—the production of antibodies [42,43,44]. The summarized results from DPV measurements carried out with the serum samples can be seen in Figure 6. Since the positive serum samples decreased the current intensity of the probe, whereas negative samples did not change the signal or slightly increased it, the data were analyzed by cut-off calculation. The cut-off was established using a 95% confidence interval for the average of normalized DPV measurements.

Based on the samples evaluated and the cut-off values obtained (95%), it was possible to identify both positive and negative samples. Therefore, human sera with a response above 95% can be considered negative since the confidence interval calculated was 102 ± 7.4%, below which the results were considered positive with a confidence interval of 78 ± 6.3%. These results highlight the ability of the immunosensor based on peptides to evaluate and distinguish between positive and negative serum samples.

## 4. Conclusions

The synthetized solid-binding peptide with affinity to graphene applied in the immunosensor effectively detected antibodies against SARS-CoV-2 (AbS). The easily synthesized peptide combines with rGO in a single step, thus facilitating the construction of the immunosensor, which is very promising in a diagnostic platform.

The immunosensor proposed in this study showed excellent selectivity for AbS, which can be attributed to the specific peptide used. This specific interaction made it possible to distinguish between SARS-CoV-2 antibodies against different viral antigens (AbS and AbN). Furthermore, the matrix effect did not interfere with the performance of the immunosensor. According to the tests against positive and negative human sera, it was possible to verify the capability of the immunosensor in determining positive and negative diagnoses of SARS-CoV-2.

Antibody detection is crucial to provide data regarding the duration of immunity in a population, assisting the public administration with knowledge of the status of infection resistance and facilitating future arrangements. The simple immunosensor platform designed in this study, along with the synthesis of specific peptides, can be easily extended to the diagnosis of infections by other viruses.

## Figures and Tables

**Figure 1 biosensors-12-00885-f001:**
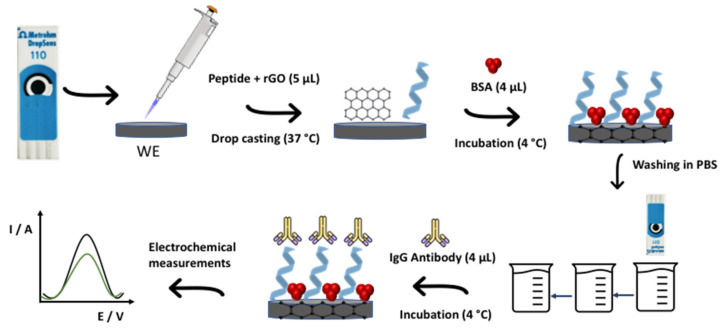
Schematic illustration of immunosensor step-by-step construction.

**Figure 2 biosensors-12-00885-f002:**
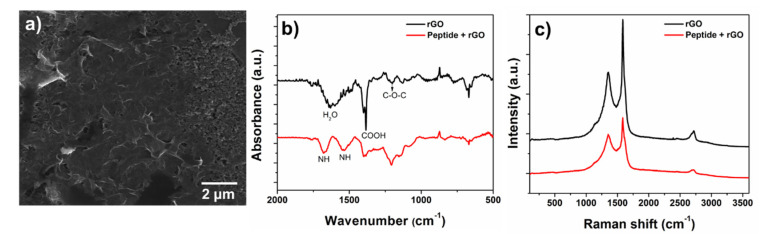
(**a**) SEM image, (**b**) FTIR spectra and (**c**) Raman spectra of rGO (black line) and mixture peptide + rGO (red line).

**Figure 3 biosensors-12-00885-f003:**
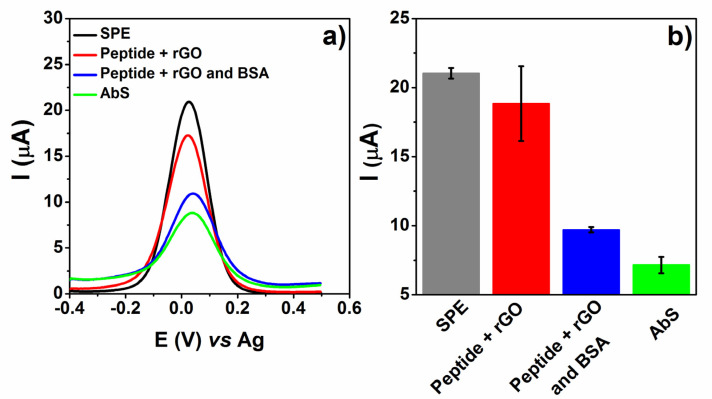
(**a**) DPVs obtained for each step of the immunosensor construction with peptide, rGO and BSA, with 1 mmol L^−1^ K_3_[Fe(CN)_6_] in PBS 0.1 mol L^−1^. (**b**) Summarized data obtained from probe current peak to each step of building up the sensor (*n* = 3, ±SD).

**Figure 4 biosensors-12-00885-f004:**
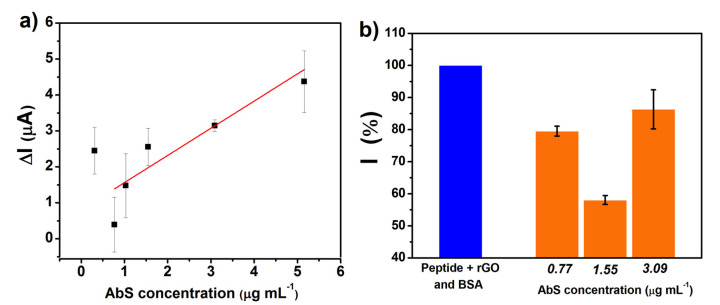
(**a**) Analytical curve obtained through probe current peak intensity vs. antibody concentration (*n* = 3, ±SD) on 0.1 mol L^−1^ PBS, pH 7.4. (**b**) Summarized DPV assays for negative human serum spiked with different AbS concentrations (*n* = 3, ± SD).

**Figure 5 biosensors-12-00885-f005:**
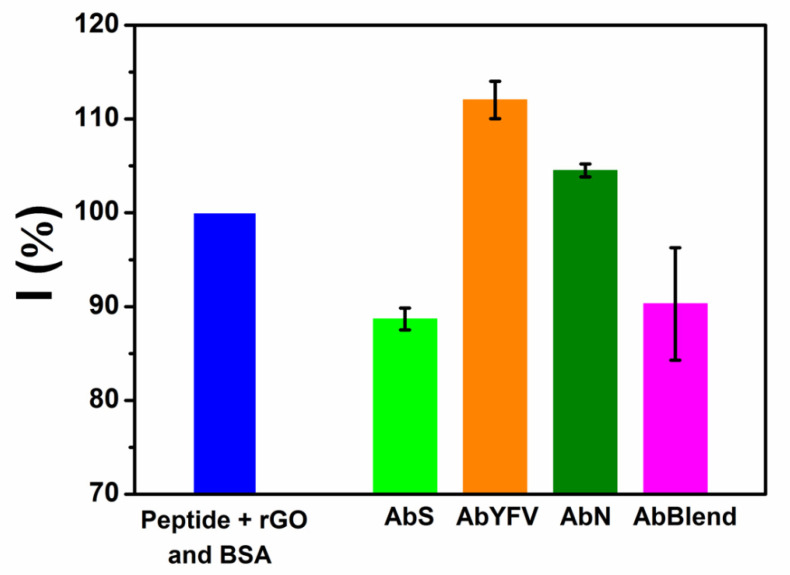
DPV data summarized for selectivity test against YFV antibodies, AbN SARS-CoV-2 and a mixture with AbS, YFV and AbN (*n* = 3, ± SD).

**Figure 6 biosensors-12-00885-f006:**
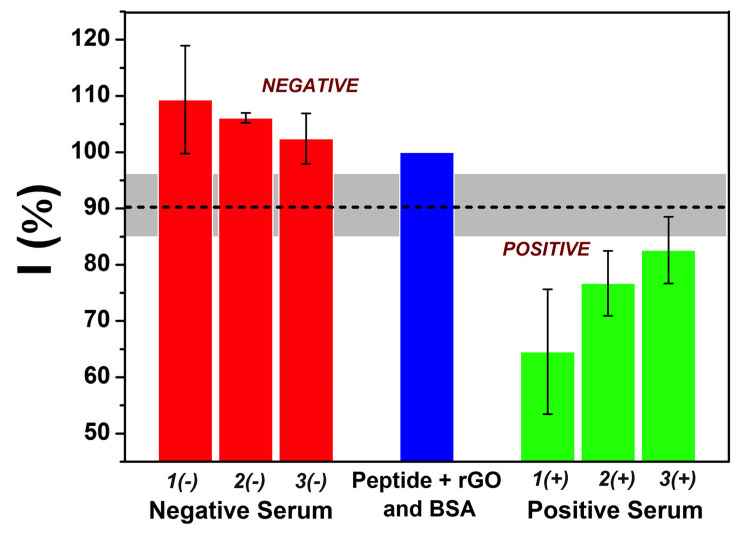
Summarized DPV data for assays with negative and positive human serum (*n* = 3, ± SD).

## Data Availability

Not applicable.

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
