# Peer review of "Graphene-Binding Peptide in Fusion with SARS-CoV-2 Antigen for Electrochemical Immunosensor Construction"

_biosensors, 2022, doi:10.3390/bios12100885_

Round 1

Reviewer 1 Report

The manuscript is interesting, but a revision is necessary.  My comments are the following:

1-Add more information to the introduction section about using peptides as elements of recognition in biosensing. Please, cite more examples that the authors have used these biomarkers

2-Please, add the functional groups to their respective values to the FTIR spectra.

3-            Perform the SEM images for the electrodes without rGO.

4-            Add a comparison table about the methods proposed for COVID-19 by using electrochemical sensors/biosensors with different forms (e.g., DNA Antibody/antigen, MIP, aptamers) to detect this disease to the Results and discussion section.

Author Response

Ms. Danna Dong

Section Managing Editor

Biosensors

Correspondence reference: biosensors-1955813

Graphene-binding peptide in fusion with SARS-CoV-2 antigen for electrochemical immunosensor construction

Dear Editor,

                Please, as previously suggested, we are sending a rewritten/revised copy of the above-cited manuscript. We would like to acknowledge the referees for their insightful suggestions and comments. The answers and comments about the points picked up are listed in the attached file and all changes made in the manuscript were highlighted in the revised version. We now believe that the manuscript is appropriate, but if you think that any changes are needed, please contact us.

Thank you very much for your attention.

Sincerely yours,

Luiz Humberto Marcolino-Junior

Reviewer 2 Report

This paper need some issues that need to be addressed

Abstract

Line 23. The authors should rewrite this sentence. The electrochemical characterization (EIS, CV, and DPV) is performed on the modified electrode not the suspension. Please verify this.

Line 29. Please check the superscript of the LOD value.

Introduction

Line 56. Reference

The authors should comment if there previous investigations, and also reference them, in which peptides binding graphene are used as immunosensors. 

Materials and Methods

Line 99. Check the subscripts of disodium hydrogen potassium formula

Line 101. Check the superscripts of the YFV concentration

Line 129. Please check the superscripts for the wavenumber of the FTIR measurements. Also check the wavenumber range is it from 4000 cm-1 or 2000 cm-1? Please verify

Line 146. As the electrode was placed using very different temperatures and even in water saturated atmosphere; the authors should clarify if there is any special cover for the electrical contacts to avoid the electrode damage.

Line 156. The authors should clarify which concentration increased? Towards which analyte? AbS?. Please rewrite because is not clear. 

Line 157. I suggest to change the concentration range from the lowest limit to the higher limit. It is a bit confusing. 

Line 157. The voltametric measurements, are CV? or DPV measurements? Or both?. Please remark this in the paragraph. 

Line 162. It is not clear what “final” concentrations means. Do the authors mean “fixed” concentration. Please, consider to change it.

Line 170. The real samples electrochemical measurements were performed using the same experimental conditions used previously? Please specify.

Results

Line 183. I understand SPE abbreviations refers to screen printed electrode, but this wasn’t specify in the text. Please verify.

Line 184, 185, 192 and 193. Please check the superscripts for the wavenumber of the FTIR measurements.

Line 195-197. Is there any evidence of this information on the literature?. The authors should confirm this data with the literature.

Line 205. Please consider to change the concentration range from the lowest to the highest concentration value. Also, it is not clear which concentration was used to evaluate the electrode suitability. Please verify.

Line 219. Please verify superscript of the AbS concentration

Line 229. Eliminate the space between K and 

Line 239. Please consider to change the concentration range from the lowest to the highest concentration value.

Line 232. Please change follow for following

Line 243. The authors should clarify how there is a lack of linearity from 0.3  testing until 5 , and they are claiming a “good linear regression coefficient R2” (0.913). This information is not clear. The authors should consider repeat this experiment and obtained a better R2 maybe in another linear range.

It would be interesting if the authors can compare their results with others reported in the literature. There is a lack of comparison with previous studies.

Author Response

(The authors gave the same response as above.)

Round 2

Reviewer 1 Report

The authors revised the manuscript, which should be accepted in Biosensors. 

Reviewer 2 Report

Thank you for incorporate the suggested changes to the submitted manuscript